# Structural Insight into the Amino Acid Environment of the Two-Domain Laccase’s Trinuclear Copper Cluster

**DOI:** 10.3390/ijms241511909

**Published:** 2023-07-25

**Authors:** Ilya Kolyadenko, Svetlana Tishchenko, Azat Gabdulkhakov

**Affiliations:** Institute of Protein Research RAS, 142290 Pushchino, Russia; sveta@vega.protres.ru (S.T.); azat@vega.protres.ru (A.G.)

**Keywords:** two-domain laccase, enzyme engineering, SgfSL, TNC, laccase trinuclear cluster, structural biology, proton transfer

## Abstract

Laccases are industrially relevant enzymes. However, their range of applications is limited by their functioning and stability. Most of the currently known laccases function in acidic conditions at temperatures below 60 °C, but two-domain laccases (2D) oxidize some substrates in alkaline conditions and above 70 °C. In this study, we aim to establish the structural factors affecting the alkaline activity of the 2D laccase from *Streptomyces griseoflavus* (SgfSL). The range of methods used allowed us to show that the alkaline activity of SgfSL is influenced by the polar residues located close to the trinuclear center (TNC). Structural and functional studies of the SgfSL mutants Met199Ala/Asp268Asn and Met199Gly/Asp268Asn revealed that the substitution Asp268Asn (11 Å from the TNC) affects the orientation of the Asn261 (the second coordination sphere of the TNC), resulting in hydrogen-bond-network reorganization, which leads to a change in the SgfSL-activity pH profile. The combination of the Met199Gly/Arg240His and Asp268Asn substitutions increased the efficiency (k_cat_/K_M_) of the 2,6-DMP oxidation by 34-fold compared with the SgfSL. Our results extend the knowledge about the structure and functioning of 2D laccases’ TNC active sites and open up new possibilities for the directed engineering of laccases.

## 1. Introduction

Laccases (EC 1.10.3.2, p-diphenol: dioxygen oxidoreductases) are among the very few enzymes that have been studied since the end of the 19th century [1,2]. These multicopper enzymes catalyze the one-electron oxidation of various aromatic compounds, coupled with the full reduction of molecular oxygen to water [3]. Laccases are characterized by their broad substrate specificity. High- and low-molecular-weight compounds of various chemicals can act as substrates: mono- and diphenols, polyphenols, diamines, aminophenols, aromatic or aliphatic amines, inorganic ions, phenolic acids, metals, and others [4,5]. Therefore, laccases are used in various industries, including the textile industry (dyes’ biodegradation) [6], medicine and healthcare (biosensors) [7], synthetic chemistry (the creation of polymers and biofuels) [8,9], ecology (the bioremediation of soil and wastewater) [10], the food industry (the extension of the shelf lives of raw materials and products), etc. [6,11]. This range of applicability contributes to their study. The engineering of their catalytic properties and methods of soft immobilization are among the main directions of research on laccases [12,13,14,15,16,17,18,19,20,21].

In general, laccases can be categorized into two subgroups based on their structural organization: three-domain laccases (hereinafter referred to as 3D) and two-domain (2D) laccases. The 3D laccases are widely distributed in nature and have been identified in higher plants, some insects, fungi, lichens, bivalves, crustaceans, bacteria, and Archaea [4,22,23,24,25,26]. The 2D laccases are produced by a small range of organisms, mainly bacteria from the genus *Streptomyces* [27,28]. Despite the differences in domain organization, the composition and geometry of the catalytic centers of 3D and 2D laccases are conservative, but their amino acid environments are varied [29]. The active centers of laccases contain four copper ions. According to their spectroscopic and paramagnetic properties, copper ions are categorized into three groups: Type-1 (one copper, T1), Type-2 (one copper, T2), and Type 3 (two coppers, T3α and T3β) [30]. 

Regarding the investigation of the catalytic mechanism of laccases [3,4,31,32,33,34,35], it should be briefly summarized that the oxidation of the substrate takes place in the substrate-binding pocket, which is mainly localized near copper T1, coordinated by three residues (two His and Cys). The electrons from the substrate are transferred via the His–Cys–His pathway to the TNC, which is formed of one T2 copper ion and two T3 copper ions (T3α and T3β). In the TNC, the reduction of O_2_ to two H_2_O occurs. This process is carried out in two stages and requires four electrons and four protons, which are transferred from T1 and charged residues, located close to the TNC. It is proposed that oxygen penetrates to the TNC via the T3 channel, whereas protons are transferred to copper ions from charged residues localized in a T2 channel. The charged residues involved in proton transfer in 3D laccases are conservative “Asp” (located in a T2 channel) and semiconservative among bacterial and fungal laccases “Asp/Glu” (located in a T3 channel). Thus, laccases catalyze two reactions at once: the oxidation of substrates in the T1 center and the reduction of oxygen in the TNC. 

The conservation of the composition and geometry of the active centers of 2D and 3D laccases defines a common mechanism in the catalyzed reaction, whereas the differences in the amino acid environment of copper ions (second coordination sphere) determine the conditions of their functioning. Structural and functional studies showed that second-coordination-sphere residues are crucial for laccase functioning because they modulate the redox potential of a T1 copper ion [4,29,36,37], as well as regulating a substrate-binding pocket geometry [12,38,39,40], the accessibility of molecular oxygen to the TNC [41], and the rate of proton transfer to the TNC copper ions [33,42,43,44]. Considering all the above, 3D laccases, especially fungal laccases, significantly exceed 2D laccases in catalytic activity, and as a result, the majority of research is aimed at engineering 3D laccases. In contrast to 3D laccases, 2D laccases are characterized by significantly higher functional stability and resistance to inhibitors [29,40,45,46,47,48]. In addition, 2D laccases can oxidize some phenolic compounds in alkaline conditions, which is not typical of most 3D laccases, as their optimum functioning and stability are in the acidic/neutral pH range [39,40,45].

Since the residues of second coordination spheres of T1 centers and TNCs in 3D and 2D laccases are various, it is known that the low activity of 2D laccase toward classical substrates (ABTS and 2,6-DMP) is due to the low redox potential of the T1 center, the inappropriate geometry of the substrate-binding pocket for bulk molecules, and the limited TNC accessibility for oxygen and protons due to the T3 and T2 channel sizes [29,38,41,42,49]. The differences in the substrate-binding regions have already been discussed by many researchers [39,50] and by us in a previous paper [40]. Apparently, these differences are related to the different functions and substrate range of 2D and 3D laccases in host organisms. The limited accessibility of the TNC might be the reason for the prevention of the penetration of inhibitors to the active center, which protects the enzyme under harsh conditions. Despite the same reaction occurring in the TNCs of 2D and 3D laccases (the reduction of oxygen to water), the TNC and amino acid environments of these enzymes are fundamentally different. The second coordination sphere of the 2D laccases’ TNCs is mostly represented by charged and polar residues, while it is mostly hydrophobic in 3D laccases [29,50].

Interestingly, the substitutions in the substrate-binding region strongly affect the pH activity profiles of some 3D laccases, while the modification/mutagenesis of the substrate-binding pockets of the 2D laccases SgfSL and *S. coelicolor* (SLAC) does not significantly affect the pH activity profile or the optimal pH (pH_opt_) [38,39,40,41,51]. We previously found that the substitution of Arg240 (located in the T2 channel) for His leads to an increase in the SgfSL activity in alkaline conditions, while combining Met199Gly (near T1 copper) and His165Ala (the second coordination sphere of T3β) substitutions leads to a sharp shift in the pH_opt_ of 2,6-DMP oxidation from pH = 8 to pH = 6.5 [40,42]. Based on the previous results, a literature review, and structural studies of 2D and 3D laccases, we suggest that the significant difference in the TNC copper environment of 2D and 3D laccases can make a major contribution to the capacity of 2D laccases to be alkaline-active towards phenolic compounds.

Our previous results showed that TNC second coordination sphere residues are not only involved in proton shuttling and regulating oxygen access, but also critical to maintaining the proper geometry of the TNC copper ions during oxygen reduction process. We have shown that the substitution of some residues near the TNC of SgfSL leads to a complete loss of the enzyme’s functional activity [41]. In light of this, and to test the hypothesis about the contribution of TNC second coordination sphere residues to 2D laccase alkaline activity, we decided to indirectly influence the amino acid environment of SgfSL TNC copper ions. For this purpose, we constructed mutant forms of SgfSL by combining previously obtained substitutions that significantly increase the catalytic activity of the enzyme (Met199Gly, Met199Ala, and Arg240His) with the substitution Asp268Asn that is outside of the TNC environment. Met199 is located in the substrate-binding pocket of SgfSL, and its replacement with Ala or Gly increases the catalytic activity of the enzyme by a factor of two or five, respectively [40]. The combination of Met199Gly substitutions with Arg240His (involved in proton transfer via the T2 channel of SgfSL) increases the efficiency of 2,6-DMP oxidation by 16-fold compared with SgfSLwt [40]. Asp268 is 11 Å from the TNC, so does not belong to the second coordination sphere of the active center and, as a result, cannot directly participate in oxygen reduction (Figure 1a). However, the side group of this residue forms a hydrogen bond (H-bond) with the NH_2_ group of Asn261 and the water molecule in the water chain that follows from the trimer central cavity to Asp260 (Figure 1a,b). Both residues belong to the second coordination sphere of T2 and T3α copper ions of SgfSL. Asp260 is an conservative analog among the various 3D laccases of the charged “Asp”, which is localized in the T2 channel and plays an important role in proton transport. The role of Asn261 in the enzymatic reaction, catalyzed by 2D laccase, is unknown as of yet. We have proposed that the substitution of Asp268 to Asn will not significantly affect the structure of the active site, but can influence Asn261 and the H-bond network, which could have an effect on SgfSL catalytic activity.

Structural and functional studies of mutant forms Met199Ala/Asp268Asn, Met199Gly/Asp268Asn, and Met199Gly/Arg240His/Asp268Asn showed that the Asp268Asn substitution, as expected, did not affect the TNC’s overall structural organization, but changed the pH optimum and, in some cases, the efficiency of 2,6-DMP oxidation. The efficiency of 2,6-DMP oxidation (k_cat_/K_M_) by the triple mutant form Met199Gly/Arg240His/Asp268Asn at pH 8.5 is about 34 s^−1^ mM^−1^, which makes it one of the most active laccases in alkaline conditions.

## 2. Results

### 2.1. SgfSL Mutant Forms’ Construction and Purification

The plasmids, carrying the SgfSL gene with the Met199Gly/Asp268Asn, Met199Ala/Asp268Asn, and Met199Gly/Arg240His/Asp268Asn substitutions, were constructed as described in Section 4.1. The quality of the obtained plasmids was assessed using electrophoretic and spectroscopic methods of nucleic acid analysis. Because of the side-chain similarity of Asp and Asn, the presence of the proper substitutions was validated not only using high-resolution X-ray analysis, but also via sequencing. The protocols of protein purification are described in Section 4.2. The quality, purity, and concentration of the mutant forms were validated using classical PAAG electrophoresis and spectroscopic methods.

### 2.2. Structural Analysis of the Mutant Enzymes 

The structures of two mutant forms, Met199Ala/Asp268Asn and Met199Gly/Arg240His/Asp268Asn, were determined. Diffraction data from crystals were collected at 120 K and showed no significant difference in the overall protein folds, but changes in the conformation of amino acid side chains were definitely detected. The structure of the Met199Ala/Asp268Asn mutant form with the resolution 2 Å demonstrated the most notable changes. The high resolution of the double mutant crystal structure allowed us to unambiguously identify the structural changes that may affect the catalytic properties of SgfSL. The most important parameters of data collection and structure refinements are given in Table 1.

### 2.3. Kinetic Analysis of the Mutant Enzymes

The catalytic activity of the mutant forms was studied using a spectroscopic method of analysis, as described in Section 4.3. All mutant forms demonstrated high enzymatic activity towards two classical substrates of laccases, ABTS and 2,6-DMP. The mutant forms of Met199Ala/Asp268Asn and Met199Gly/Asp268Asn significantly exceeded SgfSLwt in terms of the rate (k_cat_) and efficiency (k_cat_/K_M_) of ABTS oxidation. Moreover, they were not inferior to the previously obtained highly active mutant forms of Met199Ala, Met199Gly, and Met199Gly/Arg240His. At the same time, the efficiency of ABTS oxidation by the triple mutant form Met199Gly/Arg240His/Asp268Asn was slightly lower than that by Met199Gly/Arg240His, but was almost fourfold higher than that by SgfSLwt (Table 2). 

The most significant changes were observed when SgfSL mutants’ activity towards 2,6-DMP was analyzed. Whereas the pH_opt_ for ABTS oxidation between all mutant forms was similar, a significant difference in pH_opt_ for 2,6-DMP oxidation was detected (Table 2, Figure 2). The pH_opt_ values of 2,6-DMP oxidation by SgfSLwt and Met199Gly/Arg240His were almost the same and were pH = 9 and pH = 8.5, respectively (Figure 2b). Combining the Met199Ala and Asp268Asn substitutions, as well as Met199Gly and Asp268Asn, the pH_opt_ shift of 2,6-DMP oxidation to a more acidic side was detected (pH = 7 and pH = 7.5, respectively; see Table 2 and Figure 2a). A similar effect was observed when the Met199Gly and His165Ala substitutions were combined [40].

Taking into account the efficiency of 2,6-DMP oxidation, the pH_opt_ of substrate oxidation by the Met199Gly/Arg240His/Asp268Asn mutant did not change. However, if the pH_opt_ is evaluated using the rate of the catalyzed reaction, the optimal conditions for the 2,6-DMP oxidation by the triple mutant are in the pH range 7.5–8.0, which is lower than the pH_opt_ of Met199Gly/Arg240His and SgfSLwt (Table 2 and Figure 2b). It should be noted that the efficiency of 2,6-DMP oxidation by the double mutants Met199Ala/Asp268Asn and Met199Gly/Asp268Asn remained almost unchanged compared to that of the single mutants Met199Ala and Met199Gly. That was not the case with the triple mutant Met199Gly/Arg240His/Asp268Asn, where the Asp268Asn substitution doubled the efficiency of 2,6-DMP oxidation compared with Met199Gly/Arg240His and increased it more than 30-fold compared with SgfSLwt. To summarize, the Asp268Asn substitution had almost no effect on the ABTS oxidation, but significantly changed the conditions and efficiency of enzyme activity towards 2,6-DMP.

## 3. Discussion

Structural and functional studies of proteins, including industrial enzymes, are important for both the fundamental and applied science fields. Understanding the relationship between an enzyme’s structure and function makes it possible to determine the factors affecting the catalytic properties of macromolecules, which is important for the directed engineering of industrially relevant enzymes. The activity of most enzymes is achieved within a very narrow range of conditions. The main factors affecting the catalytic activity of enzymes include temperature, salt and ionic conditions, the concentration of the substrate and reaction products, and the presence/absence of activators or inhibitors [51]. By adjusting these environmental parameters, it is possible to achieve the conditions for maximum enzyme activity.

In the case of laccases, the situation becomes more complicated since these enzymes have two active centers that differ in composition and function (T1 and TNC). It has been proven that the rate of substrate oxidation by laccases depends on many biochemical and structural factors, which are detailed in Section 1. The structural features of a particular enzyme influence each of these factors. Therefore, in order to achieve the maximum catalytic activity of laccases, it is necessary to take into account a large number of different elements.

In this study, the influence of the polar residues located outside of the second coordination sphere of the TNC (Asp268) on the functioning of SgfSL was explored. Asp268 is located on the SgfSL monomer’s interface and its side chain is tightly fixed by an H-bond with the main chain of Gly265 and the side chain of Asn261, as well as electrostatically with the main chain of Ile263 (Figure 3a). The position of Asp268 (11 Å away from the TNC) does not allow this residue to be directly involved in the oxygen reduction process, but due to the environment (waters W1–W3, Figure 1 and Figure 4b), it can participate as a donor/acceptor in the transfer of protons via the Grotthuss mechanism. According to the Grotthuss mechanism, protons can be transferred between residues via the H-bond network of water molecules [52]. In the SgfSLwt structure, Asp268 bonds with the water W3 (Figure 1 and Figure 4b), which involves it in a network of hydrogen bonds and, through water molecules W1 and W2 (Figure 1 and Figure 5b), connects it with Asp260—one of the participants in the proton transfer pathway. However, Asp268 also bonds with the Asn261 side group, which is located 6.5 Å from the TNC center near the histidines, coordinating the T3α (His105 and His290) and T2 (His235) copper ions. The Asn261 side group does not interact with such histidines, but forms hydrogen bonds with Pro266 and Asp268, the roles of which in SgfSL functioning have not been investigated yet (Figure 3a). 

These hydrogen bonds fix the position of the Asn261 side group; as a result, the C=O group is turned towards the TNC copper ions (Figure 3a). Nevertheless, the C=O group of Asn261 does not have H-bonds with water molecules or histidines, coordinating TNC copper ions, and therefore cannot be involved in proton transfer, but it definitely creates electrostatic potential in the SgfSL TNC copper environment (Figure 3a). To summarize, in the SgfSLwt structure, Asp268 might be involved in proton transfer via the W1–W3 water chain localized in the channel leading from the central cavity of the SgfSL trimer to His105, and it indirectly controls the electrostatic potential of the TNC via the coordination of Asn261. 

The structural analysis of the Met199Ala/Asp268Asn mutant form showed that the Asp268Asn substitution led to the Asn261 side group conformation change. The high-resolution (2 Å) structure allowed us to detect the rotation of the NH_2_–C=O group of Asn261 by 180° and identify the double positions of the side group of this residue in some chains (Figure 3b). The NH_2_–C=O group rotation was defined by the new environment of Asn261, precisely by the NH_2_ of the Asn268 side chain, which formed a hydrogen bond with C=O of the Asn261 side chain. As a result of these changes, the NH_2_ group, rather than C=O, was localized near the TNC of the Met199Ala/Asp268Asn mutant form, which can affect the overall charge and electrostatic potential of the system. The Asn261 side group double positions were observed in the B and D chains of the Met199Ala/Asp268Asn mutant form. These conformations were stabilized by hydrogen bonds between the NH_2_ group of Asn268 and the C=O of the side group of Asn261 (Figure 3b).

Alternative conformation of the Asn261 side group changed the electrostatic potential of the system much more strongly than the turn of the side group, which could affect the TNC functioning to a greater extent. It was proven that electrostatic environmental effects contribute to the transition state energy of an enzyme’s catalyzed reactions, whereas charged amino acids close to an active center can tune the electrostatic potential [53,54]. For example, the electrostatic environment of the retinal Schiff base in bacteriotropin, especially the movement of Thr89 and the position of the side chain of Asp212, played a key role in proton transport [55].

Not only subtle side chain mobility was detected in the mutant structure. It was found that Asp268Asn substitution led to reorganizing the network of hydrogen bonds, which through water molecules W1–W3 bound Asp268 with Asp260 (Figure 4a). In the structure of the wild-type enzyme, Asn261 was unable to interact with anything but Asp268 (Figure 3a and Figure 4b). However, in the structure of the Met199Ala/Asp268Asn mutant, the geometry of the NH_2_ group of the alternative conformation of Asn261 allowed this residue to form hydrogen bonds with water molecules W2 and W3 (Figure 4a). These new hydrogen bonds allowed Asn261 to be involved not only in the creation of electrostatic potential, but also in proton transfer to the TNC via the water following from the central trimer cavity to Asp260. Most likely, the observed hydrogen bonds’ net reorganization and electrostatic potential changes of the TNC copper ions’ environment (due to NH_2_–C=O group rotation) led to the pH_opt_ shift of 2,6-DMP oxidation by the double mutants Met199Ala/Asp268Asn and Met199Gly/Asp268Asn, since other changes were not determined in the structure of the Met199Ala/Asp268Asn mutant form. At the same time, the pH_opt_ of ABTS oxidation did not change since this substrate is oxidized under proton-rich environmental conditions (pH = 4). Based on this, there arises the assumption that Asn261 and Asp268 are proton donors/acceptors and can affect the gradient of proton transfer to T2 and T3α copper ions of TNC SgfSL.

The triple mutant Met199Gly/Arg240His/Asp268Asn was obtained in order to further study the functional role of Asn261 and Asp268. The NH_2_–C=O rotation of the Asn261 group could not be accurately observed due to the relatively lower resolution of the structure. However, functional studies revealed significant changes in the oxidative activity of the triple mutant. The extended pH activity profile (from 7.5 to 8.5) of 2,6-DMP oxidation was detected (Table 2 and Figure 2b). Moreover, the rate of 2,6-DMP oxidation by the triple mutant form at pH = 7.5–8.5 was almost the same. This was not the case with the double mutant form Met199Gly/Arg240His, where the maximum rate was observed at pH = 8.5 (Figure 2b).

The expanded pH activity profile for the triple mutant may confirm Asn261 and Asp268’s involvement in the process of proton transfer to the TNC copper ions. However, the absence of a manifest shift in pH_opt_ can be explained by His240 being a stronger proton donor at pH 8.5 than Asn261 at lower pH values. Despite the fact that His cannot be a proton donor/acceptor at pH = 8.5 according to its pKa, in a recent study we showed that this particular residue is critically important for the functioning of 2D laccases in the alkaline condition [42]. Most likely, the amino acid environment of 240 residue modulates its pKa, and it obtains the ability to participate in proton transfer through the T2 channel. The efficiency of 2,6-DMP oxidation by Met199Gly/Arg240His/Asp268Asn at pH = 8.5 was twice that with the double mutant form Met199Gly/Arg240His and 30-fold higher than by SgfSLwt. This is additional evidence of Asn261 and Asp268’s important roles in the functioning of the SgfSL.

The following conclusions can be drawn from the parallel between the TNC functioning mechanisms of 2D and 3D laccases. The Glu/Asp and Asp located in the T3 and T2 channels (near the T3 and T2 copper ions) were established as the main participants in the oxygen reduction process in various 3D laccases (Figure 5a,b) [3,31,43,44,56,57,58]. Asp, located in the T2 channel near T2, provides an H-bonding network to the T2/T3β edge of the TNC, which stabilizes the coordination unsaturation in the TNC and tunes the redox properties of the T2 and T3 coppers (Figure 5a,b) [59]. It is also involved as a proton shuttle group in the oxygen reduction process. The Glu/Asp, located in the T3 channel near T3 ions, hydrogen bonds to the μ-OH bridge via crystalline water due to its involvement in the oxygen reduction mechanism as a proton donor [59]. The rest of the residues of the second coordination sphere of 3D laccase TNC are hydrophobic and form the environment of the copper ions, which creates a proton transfer gradient and promotes the entry/release of oxygen and water from the active center. The situation becomes more complicated when one takes into account the results obtained in this work. Comparing the crystal structures of 2D and 3D laccases, we noticed that the amino acid environment of the second coordination sphere of some 3D laccases is similar to that studied in this study (shown in gray and highlighted in Figure 5a,b). The identified residues and networks of hydrogen bonds can affect the functioning of enzymes in the same way as Asp268 and Asn261 in 2D laccase. It should be noted that these residues are not conservative. Nothing is known about their functional roles yet, but they can fine-tune the TNC of a particular laccase. 

Such a suggestion is in accordance with the results of the directed evolution experiments of 3D laccase from *Myceliophthora thermophila* (MtL), where the N109S mutation (second coordination sphere of T3α, PDB id 6F5K) changed the pH activity profile of 2,6-DMP oxidation from four to six [60]. As far as we know, there are only a few 3D laccases with alkaline pH_opt_ (PIE5, Pp4816, Pa5930, and MaL-M1) [61,62,63]. Pp4816 and Pa5930 are 3D laccases with very low activity from *Pediococcus pentosaceus* 4816 and *P. acidilactici* 5930. PIE5 and MaL-M1 are engineered 3D laccases from the basidiomycete *Coprinopsis cinerea* and the ascomycete *Melanocarpus albomyces*, respectively. To date, only crystal structures of low-active Pp4816 and Pa5930 have been solved, and our analysis shows that additional polar residues are located in the T2-channel—for instance, Thr420 (Pp4816, PDB id 6XJ0) in the position corresponding to the Asn261 of SgfSL. There are no structures for PIE5 and MaL-M1 so far, which makes it difficult to guess which structural elements determine the alkaline activity of PIE5 and MaL-M1. 

The situation is radically different for 2D laccase. Not only Gln292 (Glu/Asp analog located in the T3 channel) and Asp260 (Asp analog located in the T2 channel near the T2 center), but also Tyr109, Arg240, and His165 side groups (numeration of SgfSL, PDB id 6S0O, Figure 5c) take part in the reduction of oxygen to water [40,42,64,65,66]. Arg240 (T2 channel, 12Å from the T2 copper ion) and His165 (T3 channel, 4.5Å from the T3β copper ion) are proton donors/acceptors, while Tyr109 (T2 channel, 4Å from the T2 copper ion) can be an alternative electron donor in cases of T1 center functional impairment [40,42,64]. Ser294 (T3 channel, 6Å from the T3β copper ion) was recently suggested to be involved in proton transfer to TNC SLAC (Ser295 in SgfSL, PDB id 6S0O), but there is no experimental evidence to date [65]. 

Here, we demonstrated that Asp268 outside of the TNC second coordination sphere and Asn261 belonging to the second coordination sphere influence the reduction of oxygen to water by SgfSL, along with Tyr109, Arg240, His165, Gln292, and Asp260. Based on a structural and function analysis, we propose that Asp268 and Asn261 are the proton donors/acceptors that function via a newly identified channel following from the central trimer cavity of SgfSL to Asp260 and His105 (second coordination sphere of T2 and T3α). Taking into account the unexplored functional significance of many polar residues near the TNC of 2D laccases (His227, His155, Ser295, and Asp114) and those identified by the structural analysis residues and networks of hydrogen bonds, which link polar residues outside of the TNC of several 3D laccases with residues of a second coordination sphere, it can be argued that the reduction of oxygen to water by laccases is influenced by a significantly larger number of factors than were known before. The questions remain to be solved: how many channels exist for proton access in 2D and 3D laccases, and how many factors affect laccase activity?

## 4. Materials and Methods

### 4.1. Plasmid Construction 

The pET32-Xa/Lic-based plasmid, carrying the gene encoding the SgfSL_Met199Gly/Arg240His, SgfSL_Met199Gly, and SgfSL_Met199Ala mutant forms, was used as a template for the PCR with a pair of corresponding mutagenic primers (the replaced nucleotides are in italics and underlined) [67]. 

Asp268Asn_ For: 5′-ATCTGCGGCCC*A*GC*AA*ACTCCTTCGGCTTCC-3′Asp268Asn_ Rev:5′-GGAAGCCGAAGGAGT*TT*GC*T*GGGCCGCAGAT-3′

Amplification of the target template was carried out using KOD Hot Start DNA Polymerase (Novagen, Darmstadt, Germany) according to the manufacturer’s instructions and QuikChange^TM^ method protocols [68]. 

### 4.2. Purification of SgfSL Mutants 

The *E. coli* strain BL21(DE3)/Rosetta (Qiagen, Hilden, Germany) was transformed with pET32-Xa/Lic-based plasmids carrying the gene of SgfSL with different mutations (Met199Ala/Asp268Asn, Met199Gly/Asp268Asn, and Met199Gly/Arg240His/Asp268Asn). Cells were grown at 37 °C in TB (Terrific Broth) media with shaking at 160 rev/min until OD_600_ = 1. The production of mutants was induced by adding isopropyl β-D-1-thiogalactopyranoside (IPTG) to a final concentration of 0.25 mM. Along with IPTG, CuSO_4_ was added to a final concentration of 1 mM. After induction, cells were incubated for 18 h with low shaking (50 rev/min) at 25 °C. After that, cells were collected via centrifugation at 7000× *g* for 25 min, suspended in buffer A (20 mM phosphate buffer pH 7.4, containing 0.5 M NaCl and 20 mM imidazole) with 0.5 mM phenylmethylsulfonyl fluoride (PMSF) and 200 ng/mL DNaseI, and disrupted by EmulsiFlex-C3 high-pressure homogenizer (Avestin, Ottawa, ON, Canada). Cell debris was removed via centrifugation (30 min at 10,000× *g*), and the supernatant was loaded onto a column packed with Ni-NTA (GE Healthcare, Uppsala, Sweden) equilibrated with buffer A. The column was washed with buffer A, and the protein was eluted with a step gradient of buffer A with imidazole at a final concentration of 150 mM. Protein-containing fractions were collected and dialyzed against the buffer for proteolysis (50 mM Tris-HCl, pH 8.0, 100 mM NaCl). Proteolysis was carried out for 16 h at room temperature by adding 1 U of factor Xa protease (Sigma-Aldrich, Taufkirchen, Germany) per 1 mg of proteins and CaCl_2_ to 1 mM. Then, metal affinity chromatography was performed on the same column to make proteins without a thioredoxin N-terminal tail. At the final stage, proteins were concentrated to 7–20 mg/mL and dialyzed against buffer B (50 mM H_3_BO_3_–NaOH, pH 9.0, 100 mM NaCl). 

### 4.3. Kinetic Parameters of SgfSL Mutants

The laccase activities were assayed by measuring the amount of oxidized ABTS (2,2-azino-bis-(3-ethylbenzthiazoline-6-sulfonate, Sigma) or 2,6-DMP (2,6-dimethoxyphenol, Sigma) using a Cary 100 UV-Visible Spectrophotometer. The optimal pH of variants’ activities was determined at 30 °C, using the universal 50 mM Britton–Robinson (BR) buffer within the pH ranges 6.5–9.5 for 2,6-DMP and 3.0–4.5 for ABTS. The reaction mixture (1 mL) contained 1 mM of 2,6-DMP or 0.5 mM of ABTS and the enzyme variants (0.4–5 µg). The laccase activity was determined as the amount of 2,6-DMP or ABTS, oxidized at 30 °C for 1 min. The kinetic parameters were determined at the optimal pH, using a substrate range of 0.3–5 mM or 0.0625–1.5 mM for 2,6-DMP and ABTS, respectively. All experiments were performed in triplicate. The final curves fitted the Michaelis–Menten equation by nonlinear least-squares regression with SigmaPlot 11.0. When calculating the kinetic parameters, it was taken into account that the enzymes are homotrimers. Molar extinction coefficients, used to determine the maximal velocity of substrate oxidation, were ε_469_ = 49,600 M^−1^ cm^−1^ for 2,6-DMP and ε_420_ = 36,000 M^−1^ cm^−1^ for ABTS [69,70].

### 4.4. Crystallization and Crystallography 

Crystallization experiments were performed at different temperatures (23–37 °C) using the hanging-drop vapor-diffusion method on siliconized glass cover slides in Linbro plates (Molecular Dimensions, Sheffield, UK). Crystals of laccase variants for X-ray analysis were obtained using a microseeding strategy. Microcrystals of laccase mutants were grown using 20% *v*/*v* PEG Smear High, 0.1 M Bicine, pH 9.3 (condition #21 of BCS-1 from Molecular Dimensions) as precipitant. The seed stock was prepared using the original research technique [71]. Prepared microcrystal solutions were used as seed stock crystals for further crystallization trials. Crystallization drops were made by mixing 1.2 µL of protein at concentration 7–15 mg/mL, 0.3 µL of seed stock crystals, and 0.9 µL of a reservoir solution consisting of 15% *v*/*v* PEG Smear High, 0.15 M ammonium acetate, and 0.1 M sodium citrate at pH 5.0 (condition #31 of BCS-1 from Molecular Dimensions). A single crystal was flash-cooled after soaking in 20% PEG 4000, 20% glycerol, and 0.1 M Na-acetate at pH 4.5 as a cryosolution to collect the diffraction data. Diffraction data were collected using a home source Rigaku XtaLAB Synergy-S laboratory system (The Woodlands, TX, USA) [72,73,74]. Reflection data were processed and merged using CrysAlis software 42.89a [74]. The structures were determined by molecular replacement with Phaser or MolRep [75,76], using the structure of 2D laccase from *S. griseoflavus*, determined at 1.8 Å resolution (PDB id 6S0O), as a search model. Water molecules and metal ions were removed from the model. The initial model was subjected to crystallographic refinement with REFMAC5 [77]. Manual rebuilding of the model was carried out in Coot [78]. The final cycle with an occupancy refinement of the copper ions was performed in Phenix [79]. Data and refinement statistics are summarized in Table 1. The atom coordinates and structure factors were deposited in the Protein Data Bank. Figures were prepared using PyMOL [80].

## 5. Conclusions

Thanks to a complex approach, we were able to identify a new channel, which, along with the T2 channel, is involved in the saturation of the TNC of SgfSL 2D laccase with protons. Structural and functional studies of SgfSL mutants showed that even minor perturbations of the amino acid side chains of the second coordination sphere of the TNC influence laccase activity in alkaline conditions. Despite the fact that laccases have been studied for more than 100 years, many questions regarding their functioning have not yet been answered. Taking into account all the results, we suggest that the nearest amino acid environment (second coordination sphere), as well as residues indirectly affecting the TNC, must be considered for successful engineering of the catalytic properties of laccases. From our point of view, this study opens up new possibilities for the directed engineering of laccases. 

## Figures and Tables

**Figure 1 ijms-24-11909-f001:**
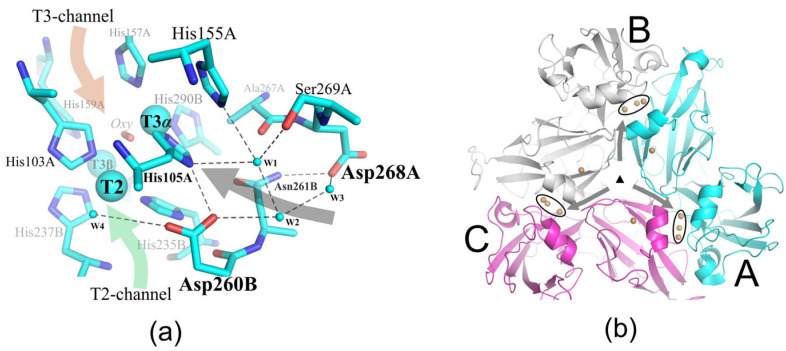
(**a**) The position and contacts of Asn268 in the SgfSLwt structure (PDB id 6S0O). Amino acid residues are shown in cyan. Copper ions are indicated as large cyan spheres. Dark dashed lines show hydrogen bonds. Small cyan spheres represent water molecules. The green and red arrows denote T2 and T3 channels, respectively; the dark gray arrow shows the proposed additional channel leading from the central cavity of SgfSL to TNC. (**b**) The trimer of SgfSL. Copper ions are shown as brown spheres and TNC copper ions are circled. The triangle in the middle represents the symmetry operator for trimer, whereas the arrows stand for the proposed route of the protons from the central cavity of SgfSL to TNC. A, B, and C in (**b**) denote the different chains.

**Figure 2 ijms-24-11909-f002:**
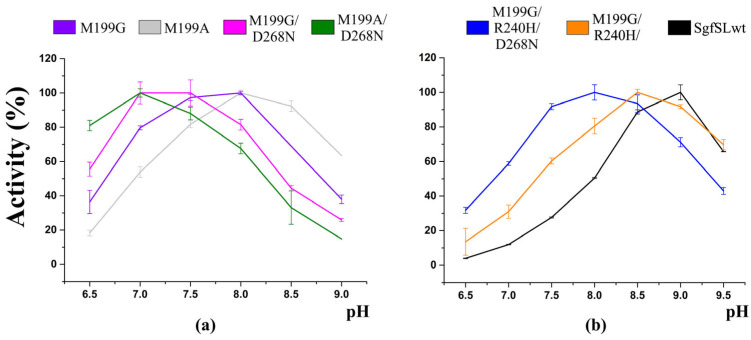
The effect of pH on laccase activity with 2,6-DMP as a substrate. (**a**) pH activity profile of M199G, M199A, M199G/D268N, and M199A/D268N mutant forms. (**b**) pH activity profile of SgfSLwt, M199G/R240H/D268N, and M199G/R240H mutant forms. The colors of the curves correspond to the colors from the plot legend. The data for the SgfSLwt, M199G, M199A, and M199G/R240H pH activity profile are taken from our previously published work [40].

**Figure 3 ijms-24-11909-f003:**
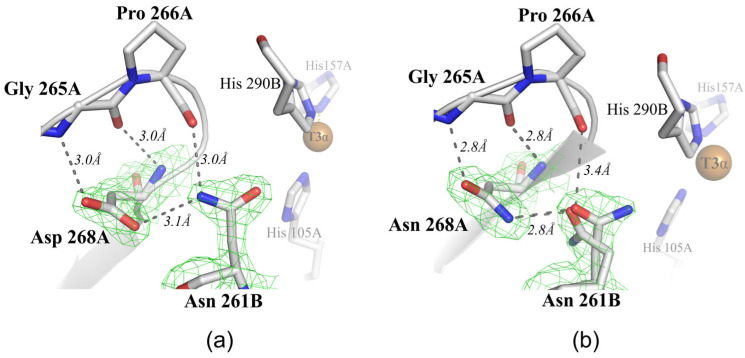
The position of Asn261 and Asp268 and its contacts in the structures of the (**a**) SgfSLwt (PDB id 6S0O) and (**b**) Met199Ala/Asp268Asn mutant (PDB id 8P9U). Amino acid residues are depicted in gray. Copper ions are shown as brown spheres. Dark dashed lines stand for hydrogen bonds fixing the position of the Asn261 and Asp268 side groups. The oxygen ligand is shown with a red stick and labeled “Oxy”. A and B in the residue names denote chains. Green mesh shows the electron density (2F_o_-F_c_, σ = 1.8) of Asn261 and Asn268.

**Figure 4 ijms-24-11909-f004:**
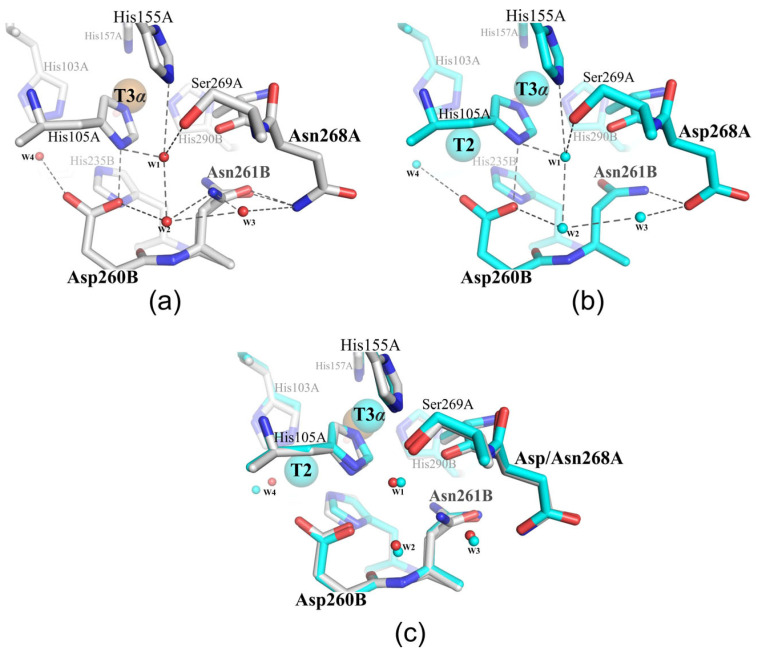
The hydrogen bond network in Met199Ala/Asp268Asn (**a**) and SgfSLwt (**b**). (**c**) The superposition of the structures. Amino acid residues are shown in gray (**a**,**c**) and cyan (**b**,**c**) and highlighted. Copper ions are depicted as large brown (**a**,**c**) and cyan (**b**,**c**) spheres. Dark dashed lines indicate hydrogen bonds. Small red (**a**,**c**) and cyan (**b**,**c**) spheres represent water molecules. A and B in the residue names stand for the different chains.

**Figure 5 ijms-24-11909-f005:**
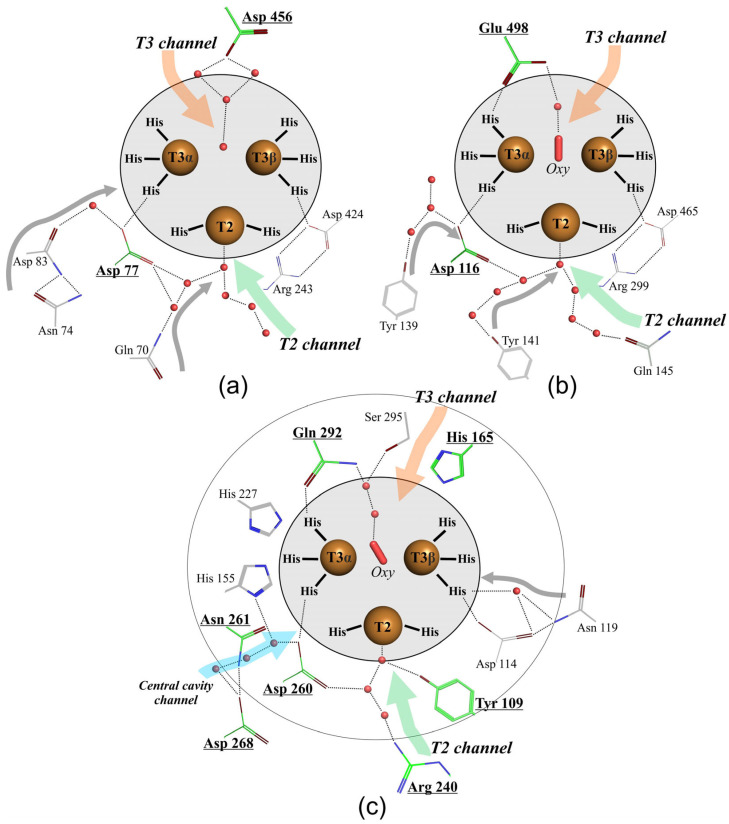
Schematic representation of the TNC amino acid environment of: (**a**) fungal 3D *Trametes maxima* laccase (PDB id 2H5U), (**b**) bacterial 3D *Bacillus subtilis* laccase (PDB id 1W6L), and (**c**) 2D laccase SgfSL (PDB id 6S0O). Copper ions are shown as brown spheres. Amino acids of the first coordination sphere of a TNC are circled in gray and called “His”. Charged amino acids of the second coordination sphere involved in oxygen reduction are shown as green lines (**a**–**c**) and circled (**c**), in bold, and underlined. The oxygen ligand is shown as a red bold stick and called “Oxy”. Water molecules are depicted as small red spheres. Amino acids, which probably influence laccase activity, are given in gray. T2 and T3 channels are depicted as pale red and green arrows and are in italics, and the central cavity channel is shown in pale cyan. Proposed proton routes are shown with gray arrows.

**Table 1 ijms-24-11909-t001:** Crystallographic data collection and refinement statistics.

	M199A/D268N	M199G/R240H/D268N
**Data collection**		
**Wavelength (Å)**	1.54	1.54
**Resolution range (Å)**	23.54–2.00 (2.07–2.00) ^a^	23.70–2.20 (2.28–2.20) ^a^
**Space group**	*P2_1_*	*P2_1_*
**Cell parameters** a,b,c (Å) α = γ = 90, β (°)	74.30, 94.02, 118.92 91.10	74.36, 93.91, 119.59 91.10
**Collection temperature (K)**	120	120
**Total reflections**	299,870 (30,309)	376,173 (38,889)
**Unique reflections**	104,468 (10,454)	82,258 (8,233)
**Multiplicity**	2.9 (2.9)	4.6 (4.7)
**Completeness (%)**	94.60 (95.30)	98.40 (96.66)
**Mean I/sigma (I)**	8.01 (1.43)	4.89 (1.35)
**Wilson B-factor (Å^2^)**	21.5	17.8
**CC_1/2_**	0.98 (0.54)	0.96 (0.49)
**Refinement**		
**Resolution range**	23.54–2.00 (2.02–2.00)	23.70–2.20 (2.23–2.20)
**Reflections used in refinement**	104,300 (3,273)	82,065 (2,593)
**Reflections used for R-free**	5231 (161)	4124 (139)
**R-work, %**	20.11 (35.49)	21.49 (27.12)
**R-free, %**	24.18 (37.39)	25.81 (32.68)
**RMSD bond lengths (Å)**	0.008	0.009
**RMSD bond angles (◦)**	0.89	0.99
**Ramachandran favored (%)**	96.61	96.66
**Ramachandran allowed (%)**	3.33	3.28
**Ramachandran outliers (%)**	0.06	0.06
**Average B-factor (Å^2^)**	27.42	20.69
macromolecules	27.49	20.73
ligands	29.76	23.25
solvent	26.03	15.48
**PDB ID**	8P9U	8P9V

^a^ Values in parentheses are for the highest-resolution shell.

**Table 2 ijms-24-11909-t002:** Kinetic constants and pH_opt_ of ABTS and 2,6-DMP oxidation by mutant forms and SgfSLwt.

Substrate	Object	pH	K_M_ (mM)	Kcat (sec^−1^)	kcat/K_M_ (sec^−1^ mM^−1^)
**ABTS**	SgfSLwt *	4.0	0.36 ± 0.07	15.11 ± 1.07	41.98
	Met199Ala/Asp268Asn	4.0	0.11 ± 0.03	21.41 ± 1.12	194.62
	Met199Gly/ Asp268Asn	4.0	0.22 ± 0.03	37.70 ± 2.41	171.40
	Met199Gly/Arg240His/ Asp268Asn	4.0	0.25 ± 0.02	38.90 ± 0.99	155.65
**2,6-DMP**	SgfSLwt *	9.0	0.32 ± 0.08	0.35 ± 0.02	1.09
	Met199Ala/Asp268Asn	7.0	0.87 ± 0.1	0.80 ± 0.03	0.92
	Met199Gly/ Asp268Asn	7.5	0.29 ± 0.02	1.88 ± 0.05	6.49
	Met199Gly/Arg240His/ Asp268Asn	7.5	1.16 ± 0.17	12.84 ± 0.65	11.07
		8.0	0.50 ± 0.08	11.10 ± 0.45	22.19
		8.5	0.24 ± 0.03	8.19 ± 0.13	34.14

* The data were copied from our previously published work [40].

## Data Availability

MDPI Research Data Policies.

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
