# Peer review of "Structural Insight into the Amino Acid Environment of the Two-Domain Laccase’s Trinuclear Copper Cluster"

_ijms, 2023, doi:10.3390/ijms241511909_

Round 1

Reviewer 1 Report

The manuscript described two mutants of the two-domain Laccase SgfSL with crystals structures and increased enzymatic activities. Though the author determined the structures of the two mutants, there were no major difference with the wild type, and the reasons for selecting the mutants were not clearly described by author. In addition to the lack of the innovation, the data of the manuscript is also insufficient, with Figure 2 is not directly related to the main text of the article.

Some issues are listed as below

1.       Section 2.1 was Not part of the results, which should be placed in methods.

2.       There are many non-standard writing issues in the article, such as the Space group of P should be italic written. Letter k of the kcat should be lowercase written, and so on.

3.       The CC1/2 and Mean I/sigma(I) value seem not matched with the resolution.

4.       The 2Fo-Fc or Fo-Fc map of the mutant residues should be provided.

5.       The conclusion was not well written.

Author Response

Dear Reviewer 1, we thank you for your comments and tried to improve the quality of the article. Also, we have carefully described the reasons for selecting the mutants and redesigned the logic of the article. All points that have been added or changed are marked in yellow.

  1. Section 2.1 was Not part of the results, which should be placed in methods.
  • We have a little changed this part and added additional information
  1. There are many non-standard writing issues in the article, such as the Space group of P should be italic written. Letter k of the kcat should be lowercase written, and so on.
  • We have carefully reviewed the article and tried to reduce the number of mistakes and non-standard writing issues
  1. The CC1/2 and Mean I/sigma(I) value seem not matched with the resolution.
  • According to the literature, data below, CC1/2 = 0.3 shall be discarded [Luo, Z.; Rajashankar, K.; Dauter, Z]. Weak data do not make a free lunch, only a cheap meal [Acta Crystallogr. Sect. D Biol. Crystallogr. 2014, 70, 253–260; Karplus, P.A.; Diederichs, K.] Assessing and maximizing data quality in macromolecular crystallography. [Curr. Opin. Struct. Biol. 2015, 34, 60–68]. Our experience shows, depending on data quality, the useful range of CC1/2 lies between 1 and 0.5.
  1. The 2Fo-Fc or Fo-Fc map of the mutant residues should be provided.
  • We have provided an 2Fo-Fc map for SgfSLwt and Met199Ala/Asp268Asn (Figure 3a and 3b)
  1. The conclusion was not well written.
  • We have tried to improve the conclusions

Reviewer 2 Report

In their manuscript, Kolyadenko et al. present biochemical and structural data of three engineered variants of the two-domain laccase SgfSL from Streptomycin griseoflavus. The work presented expands on previous published research by the authors, aimed at improving the catalytic properties of SgfSL and at understanding how the environments of the copper ions at the trinuclear center and substrate binding site affect catalytic efficiency and pH-activity optimum. Essentially, the authors discuss the effects of combining a new single amino acid mutation (Asp268Asn) with three mutants previously analysed and published (Met199Ala, Met199G and Met199G/Arg240His). Compared to their parent mutants, the combined mutants display a lowering in pH-activity optimum of 0.5-1 units, and, only for the triple mutant (Met199G/Arg240His/Asp268Asn), a significant improvement in catalytic efficiency for 2,6-DMP oxidation. Crystal structures of the M199A/D268N and M199G/R240H/D268N mutants give some clues how the D268N mutation may affect the pH optimum for 2,6-DMP oxidation, by altering the charge of the TNC environment and/or by affecting a putative proton-relay network required for the oxidation of oxygen to water.  Compared to previous published work by the authors (ref. 41 and 51), the scope of the current manuscript and gained insights are limited. Still, the work described has some merit in that the results augment the structure-function data on 2D laccases, which are less well studied than 3D laccases.

In my view, the manuscript suffers from several inconsistencies which need clarification and improvement.

Main comment: 

In the abstract (lines 13-16) it is stated that the authors show that the unique pH spectrum (pH-activity profile) of SgfSL is determined by the charged residues of the second coordination sphere of the TNC. There are two problems with this statement: (i) the manuscript only describes the effect of a single mutation in the environment of the TNC (Asp268Asn), and according to Figure 4  the mutated residue is not part of the second coordination sphere. (ii) at pH > 7, and according to Fig. 4, the charged residues in the second coordination sphere are Asp114 and Asp260 (assuming that the pKa’s of His155, His165 and His227 are ~6). None of these residues are mutated in the present study, so it is unclear to me how the authors can make their claim. I think what is intriguing in this study is how a mutation just outside the second coordination sphere can have such a significant effect on the pH-activity profile of the enzyme. This aspect needs a better discussion and explanation than currently provided by the authors.

Additional comments:

Throughout the manuscript the term “pH spectrum”  (or spectra) is used. I think this should be changed to “pH-activity profile”

Also throughout the manuscript: Kcat should be changed to kcat 

Abstract, lines 22-22: “Our results provide a compelling answer to the long-standing question regarding the unique structure of the active site of 2D laccases …”  The nature of both question and answer is not clearly defined in the abstract or manuscript.

page 2, lines 60-62: “Thus, laccases catalyse two reactions at once ….” Is this true? I would argue that the two reactions cannot occur concomitantly, but proceed sequentially since an electron transfer is involved.

page 2, lines 84-85: “The second coordination sphere of the TNC of 2D laccases is mostly represented by charged residues …”  As already discussed, this statement is debatable. Al low pH it is true that the majority of residues in the second coordination sphere is charged, as the three histidines will then be fully protonated. But at higher pH (pH > 7) the histidines are neutral (unless their pKa’s are significantly shifted), so only two out 9 residues are charged (Figure 4C).    I would change “charged residues” to “charged or polar residues”

page 3, line 130: “spectral methods” should be “spectroscopic methods”

page 3, line 132: “controlled” should be “validated”

page 4, lines 144-145: “The high resolution of the crystal structure …… on the structure and function of SgfSL”  This sentence is unclear and should be revised. I assume the authors want to emphasise that the structure of the double mutant is accurately determined allowing unambiguous identification of the structural changes that may affect the catalytic properties of SgfSL.

Table 1: Units are missing for wavelength, resolution range, Wilson B-factor, RMSD (bond lengths and angles), Average B-factor

RMS should be RMSD, bonds should be bond lengths, angles should be bond angles

Table 2: the kinetic data for SgfSLwt appear to be copied from previous published work (reference 41). This should be mentioned in the table.

Page 7, Figure 2: Similarly, the pH-activity profiles of M199G, M199A, M199G/R240H and SgfSLwt appear to be copied from reference 41. This should be mentioned in the legend.

page 7, lines 212-220, page 8, lines 228-229: “In the work, the influence of the polar residues in the second coordination sphere os TNC (Asn61) on the functioning of SgfSL was studied”. This is misleading. The work studies the effect of mutating an acidic, negatively charged residue bordering the second coordination sphere of TNC (Asp268Asn) on the functioning of SgfSL. The effect may be directly related to the Asp268Asn mutation due to the change in charge, and indirectly due to inducing an altered side chain conformation of Asn261 in the second coordination sphere of TNC. It’s unlikely, though,  that the side chain rotation of Asn261 will significantly affect the charge distributions in the environment of the active site. Rather it affects the hydrogen bonding network to which this residue participates, and the observed catalytic effects could therefore indicate that this residue plays a role in proton transfer to the active site. 

page 8, lines 238-240: “In this case, the environment of the TNC cluster looses both positive and negative charge …”  There is no loss of positive charge. The rotations of the Asn261 side chain perhaps lead to a change in electrostatic potential at the active site environment, but I expect that this change is minimal. The effect on hydrogen bonding networks will be more substantial.

page 8, lines 257-259: “But the absence of a manifest shift in pHopt can be explained by His240 as a stronger proton donor at pH 8.5 than Asn261 at lower pH values”. This sentence is very unclear and needs clarification. His240 cannot act as a proton donor at pH 8.5 (it will be mainly present as base).  Asn261 also cannot act as proton donor. Both residues could act in relaying a proton though. 

Figure 4C: the side chain of Asn261 should be flipped

page 12, line 400: “oxidoreductases” should be changed to “laccases”

I recommend to have the manuscript checked by a native speaker, as there are several grammatical errors or wrong uses of words which may lead to misinterpretations. 

Author Response

Dear Reviewer 2, we are very grateful for your comments. We agree with almost all of your comments and remarks. We carefully redesigned the logic of the article based on your comments, and tried to correct errors and omissions. All points that have been added or changed are marked in yellow.

The answers for the main comments:

“In the abstract (lines 13-16) it is stated that the authors show that the unique pH spectrum (pH-activity profile) of SgfSL is determined by the charged residues of the second coordination sphere of the TNC. There are two problems with this statement:

  • The manuscript only describes the effect of a single mutation in the environment of the TNC (Asp268Asn), and according to Figure 4 the mutated residue is not part of the second coordination sphere.
  • We have changed the phrase in the lines 13-16, to avoid the misunderstanding and reduce the categoricalness of the assumption.
  • at pH > 7, and according to Fig. 4, the charged residues in the second coordination sphere are Asp114 and Asp260 (assuming that the pKa’s of His155, His165 and His227 are ~6).”.
  • The values for His pKa’s are true in theory and for isolate amino acids, but not always valid for His’s and other charged residues buried in a proteins core and involved the enzyme functioning. It is shown that amino acids pKa might be modulated by environment [10.1016/j.bpj.2020.02.027, 10.1002/prot.10153]. So, at pH >7 not only Asp114 and Asp260 might be charged. Such assumption is supporting by our previous work, when replacement of Arg240 to His seriously changed activity of SgfSL at alkaline condition [10.1080/07391102.2021.1911852].

None of these residues are mutated in the present study, so it is unclear to me how the authors can make their claim.

  • That’s is true, here we just wanted to emphasize that the 2D laccase activity influenced not only by 2nd coordination sphere residues, but also with remote residues. The rest of the polar/charged residues close to SgfSL TNC copper ions will be investigated in our future articles. We have redesigned the article to avoid the misunderstanding and reduce the categoricalness of the claim.

I think what is intriguing in this study is how a mutation just outside the second coordination sphere can have such a significant effect on the pH-activity profile of the enzyme. This aspect needs a better discussion and explanation than currently provided by the authors.

  • I would like to say that in the revised version of the article we tried to clearly describe the principle that guided us when choosing mutations. And we tried to better discuss the effect of the mutations on the functional and structural levels.

The answers for the Additional comments:

Throughout the manuscript the term “pH spectrum”  (or spectra) is used. I think this should be changed to “pH-activity profile”

  • In almost all cases, we have replaced “pH spectrum” to “pH-activity profile”

Also throughout the manuscript: Kcat should be changed to kcat 

  • We have changed this mistake

Abstract, lines 22-22: “Our results provide a compelling answer to the long-standing question regarding the unique structure of the active site of 2D laccases …”  The nature of both question and answer is not clearly defined in the abstract or manuscript.

  • The lines 22-22 were rewrite to “Our results extend the knowledge about the structure and functioning of 2D laccases TNC active site and open up new possibilities for directed engineering of laccases..”

page 2, lines 60-62: “Thus, laccases catalyse two reactions at once ….” Is this true? I would argue that the two reactions cannot occur concomitantly, but proceed sequentially since an electron transfer is involved.

  • That’s is mistake. We have changed to “These multicopper enzymes catalyze one-electron oxidation of various aromatic compounds, coupled with full reduction of molecular oxygen to water”

page 2, lines 84-85: “The second coordination sphere of the TNC of 2D laccases is mostly represented by charged residues …”  As already discussed, this statement is debatable. Al low pH it is true that the majority of residues in the second coordination sphere is charged, as the three histidines will then be fully protonated. But at higher pH (pH > 7) the histidines are neutral (unless their pKa’s are significantly shifted), so only two out 9 residues are charged (Figure 4C). I would change “charged residues” to “charged or polar residues”

  • We have discussed our vision about charge of the residues close to the active center in the previous answer. But to avoid misunderstanding, we have changed “charged residues” to “charged and polar residues”

page 3, line 130: “spectral methods” should be “spectroscopic methods”

  • We have changed this point (line 157)

page 3, line 132: “controlled” should be “validated”

  • We have changed this term (line 159)

page 4, lines 144-145: “The high resolution of the crystal structure …… on the structure and function of SgfSL”  This sentence is unclear and should be revised. I assume the authors want to emphasise that the structure of the double mutant is accurately determined allowing unambiguous identification of the structural changes that may affect the catalytic properties of SgfSL.

  • Thank you so much for this comment. We have corrected this suggestion and have used your option (lines 171-173).

Table 1: Units are missing for wavelength, resolution range, Wilson B-factor, RMSD (bond lengths and angles), Average B-factor, RMS should be RMSD, bonds should be bond lengths, angles should be bond angles

  • We have added additional information and corrected the points you mentioned

Table 2: the kinetic data for SgfSLwt appear to be copied from previous published work (reference 41). This should be mentioned in the table.

  • The relevant reference was added to the table 1 (line 193)

Page 7, Figure 2: Similarly, the pH-activity profiles of M199G, M199A, M199G/R240H and SgfSLwt appear to be copied from reference 41. This should be mentioned in the legend.

  • The correspond reference was added to the figure 2 (lines 220-221)

page 7, lines 212-220, page 8, lines 228-229: “In the work, the influence of the polar residues in the second coordination sphere os TNC (Asn61) on the functioning of SgfSL was studied”. This is misleading. The work studies the effect of mutating an acidic, negatively charged residue bordering the second coordination sphere of TNC (Asp268Asn) on the functioning of SgfSL. The effect may be directly related to the Asp268Asn mutation due to the change in charge, and indirectly due to inducing an altered side chain conformation of Asn261 in the second coordination sphere of TNC. It’s unlikely, though, that the side chain rotation of Asn261 will significantly affect the charge distributions in the environment of the active site. Rather it affects the hydrogen bonding network to which this residue participates, and the observed catalytic effects could therefore indicate that this residue plays a role in proton transfer to the active site. 

  • Thank you for the comment. We carefully analyzed the obtained structures and found that the Asp268Asn substitution not only leads to a change the conformation of the Asn261 side group, but also affects the network of hydrogen bonds near Asp260- one of the most important residues for the SgfSL functioning. In the revised version we tried to carefully describe the changes taken away, and suggested how they can affect the SgfSL functioning.

page 8, lines 238-240: “In this case, the environment of the TNC cluster looses both positive and negative charge …”  There is no loss of positive charge. The rotations of the Asn261 side chain perhaps lead to a change in electrostatic potential at the active site environment, but I expect that this change is minimal. The effect on hydrogen bonding networks will be more substantial.

  • We agree with your comment. We have rewritten this part according to your suggestion

page 8, lines 257-259: “But the absence of a manifest shift in pHopt can be explained by His240 as a stronger proton donor at pH 8.5 than Asn261 at lower pH values”. This sentence is very unclear and needs clarification. His240 cannot act as a proton donor at pH 8.5 (it will be mainly present as base).  Asn261 also cannot act as proton donor.

  • In order to clarify our position regarding the possibility of the participation of His240 and Asn261 in the transfer of protons, we have added a few sentences to the body of the article (lines 330-334). We consider that the amino acid environment can modulate the pKa of the particular residues. We also agree that "both residues could act in relaying a proton though" but this assumption didn’t describe why replacement of Arg240 to His increase the activity of SgfSL.
  •  

Figure 4C: the side chain of Asn261 should be flipped

  • The side chain of Asn261 on Figure 4C (Figure 5C in the revised version) was flipped

page 12, line 400: “oxidoreductases” should be changed to “laccases”

  • We rewrote the conclusion and corrected this moment

Round 2

Reviewer 1 Report

The manscript entitled "Structural Insight into the Amino Acid Environment of the Trinuclear Copper Cluster of Two-domain Laccases " has been revised carefully and can be accepted in current form.

Reviewer 2 Report

This revision has significantly improved the manuscript. The authors have satisfactory addressed the comments and concerns made by the reviewers. I have no further comments, and recommend the manuscript for publication.